# Factors Influencing Problem-Solving Competence of Nursing Students: A Cross-Sectional Study

**DOI:** 10.3390/healthcare10071184

**Published:** 2022-06-24

**Authors:** Eunhee Choi, Jaehee Jeon

**Affiliations:** 1Department of Nursing, Korean Bible University, 32 Dongil-ro(st) 214-gil, Nowon-gu, Seoul 01757, Korea; ichoi9201@naver.com; 2Department of Nursing, Gangneung-Wonju National University, Gangneung-si 26403, Korea

**Keywords:** nursing, students, problem-solving, metacognition, communication

## Abstract

Problem-solving ability is an important competency for nursing students to enable them to solve various problems that occur in dynamic clinical settings. The purpose of this cross-sectional study was to identify the factors that affect the problem-solving ability of nursing students. The subjects of this study were 192 nursing college students in their second year or beyond. The research tool consisted of an online questionnaire, with a total of 91 items regarding general characteristics, metacognition, and communication competence. Data collection was conducted from 10 to 30 March 2022. An online survey link was uploaded to the student group of a social network service from two nursing colleges that permitted data collection. Subjects who agreed to participate directly accessed and responded to the online survey. The collected data were analyzed using descriptive statistics, and the factors associated with the problem-solving ability of nursing students were examined using hierarchical multiple regression analysis. The subjects’ mean problem-solving ability score was 3.63 out of 5. Factors affecting problem-solving ability were age, communication competence, and metacognition, among which metacognition had the greatest influence. These variables explained 51.2% of the problem-solving ability of nursing students. Thus, it is necessary to provide guidance to improve metacognition and to develop educational methods to improve communication competence in curricular and non-curricular courses to improve the problem-solving ability of nursing students.

## 1. Introduction

Nurses must have the ability to develop individual problem-solving methods to satisfy their patients’ diverse and high-level health needs [1]. However, the medical field is characterized by uncertainty, instability, and unpredictability; thus, it is not easy for nurses to apply or utilize the knowledge learned within controlled situations in schools, and therefore, it is often difficult for nurses to address the health needs of patients [2]. The problem-solving ability required in such situations is an essential skill that college students majoring in nursing must have in order to effectively perform their assigned nursing tasks after graduation, while successfully adapting in order to practice in a rapidly changing medical field [3]. Therefore, strategies to improve the problem-solving ability of nursing students should be applied in their education.

This study aimed to examine the relationship between metacognition, communication competency, and the problem-solving ability of nursing students, and to identify factors that affect nursing students’ problem-solving ability. The results represent basic data that could help inform the development of educational strategies to improve the communication skills of nursing students.

### Background

Problem solving involves recognizing the difference between the problem solver’s current state and the goal state to be reached, and resolving the obstacles that prevent them from achieving the goal [4]. Acquiring problem-solving ability based on judgment and critical thinking is an important element of nursing education [5]. Furthermore, the use of effective problem-solving strategies based on professional knowledge is a competency that a professional nurse should possess [3].

Recently, metacognition has been recognized in psychology and pedagogy as a central element of the understanding, self-learning, communication, and problem-solving processes. Metacognition is the ability to think about one’s own thinking [6], as expressed by the individual knowing and controlling their thought processes and applying previously acquired knowledge, skills, and experiences using appropriate strategies [7]. In particular, metacognition is an important variable for learning and problem-solving. It is the knowledge related to the selection of an appropriate strategy for the task; the establishment, selection, and application of problem-solving measures; evaluation of the effectiveness of the applied measures; and checking and adjusting the performance process [6,7]. Accordingly, interest in metacognition is increasing in the field of nursing, with emphasis on the problem-solving ability of nursing students [8]. 

Metacognition affects problem solving by cultivating the learner’s active attitude, linking existing knowledge with new knowledge, and fostering the development of practical cognitive strategies that can be used for problem solving [7]. Previous studies on learners’ metacognition have observed that a higher level of metacognition leads to an improvement in the problem-solving process, as metacognition has a significant effect on goal setting and problem-solving performance [9]. Thus, metacognition and problem-solving ability are closely related; furthermore, metacognition is a key strategic aspect in the problem-solving process [10]. Although metacognition is a powerful predictor of learning outcomes and problem-solving success, it is not clear how metacognition works regarding cognitive strategies and learning outcomes [11]. A study of adolescent metacognition-related cognition (learning strategies and problem-solving strategies) and how metacognition affects various types of learning performance confirmed that problem solving is the only mediator between general metacognition and learning performance [11]. Metacognition plays a major role in improving learning and work ability, and the appropriate use of metacognition when performing nursing tasks can improve the personal lives of nurses [12]. Therefore, it is necessary to assess the influence of metacognition and problem-solving ability on nursing education.

Communication competence is a major factor that affects problem-solving ability [13]. It is essential for smoothly maintaining professional relationships with medical personnel, patients, and guardians in diverse and complex medical environments [14]. Since communication competence is an important aspect of problem solving, it can affect the quality of nursing and the satisfaction of the patient [15]. There is an urgent need to improve communication competence, as various issues that arise during the clinical process can be solved through communication between medical staff and patients, and nurses are responsible for much of the continuous communication with patients and their guardians [16]. However, a lack of communication competence among nurses may lead to miscommunication, and thereby, poor outcomes, even if nurses possess good problem-solving methods [13].

While previous studies have revealed the relationship between communication competence and the problem-solving ability of nursing students [13,17], there is a lack of research regarding the relationship between metacognition, communication competence, and problem-solving ability. Through this research, we confirmed the relationship between the metacognition and problem-solving ability of nursing college students, the relationship between communication ability and problem-solving ability, and finally, examined the factors that affect variables associated with problem-solving ability, including metacognition.

## 2. Materials and Methods

### 2.1. Design

This study used quantitative methods to investigate the relationship between the metacognition, communication ability, and problem-solving ability of nursing students and to identify factors that affect problem-solving ability.

### 2.2. Participants and Procedures

The subjects of this study were nursing students enrolled in two nursing colleges in the same city and region. Since data collection was conducted in March, second-year, third-year, and fourth-year students, with college life experience, were targeted. The required number of study subjects was calculated using the G-Power 3.1.9.7 program, which determined a significance level of 0.05, an effect size of 0.15, a power of 0.90, and 11 predictors, for multiple regression analysis. The minimum sample size was 152. In consideration of the dropout rate, 170 was set as the target number of subjects.

Data collection was conducted from 10 to 30 March 2022. The survey was conducted online. Consent for participation in the study was obtained by the individual reading the explanation of the purpose of the study and checking a consent box, which was displayed on the first screen of the online survey. The study subjects were redirected to the questionnaire completion page after providing their consent. It required approximately 10 min to complete the questionnaire. A total of 200 participants accessed and completed the questionnaire; 192 questionnaires were used for analysis after excluding 8 questionnaires that were determined to have been completedinccurately.

### 2.3. Instruments

The tools of this study consisted of a total of 91 items, including 9 items regarding general characteristics, 20 metacognition items, 15 items dealing with communication competencies, and 45 regarding problem-solving skills.

The items regarding the general characteristics of the subjects included age, sex, academic level, interpersonal relationships, satisfaction with major, problem based learning (PBL) experience, number of related experiences, clinical practice experience (yes or no), and number of weeks of clinical practice experience.

Metacognition was assessed using the state metacognitive inventory developed by O’Neil Jr et al. [18], as modified and supplemented by Joo [19]. It consists of a total of 20 items that assess the four sub-factors of metacognition: cognition, cognitive strategy, plan, and monitoring, using a 5-point Likert scale ranging from 1 point, for ‘not at all’, to 5 points, for ‘strongly agree’. The Cronbach’s α measure of the reliability of the tool was 0.86 at the time of development and 0.89 in the study of Joo [19]. The value in the current study was 0.91.

Communication competence was measured using a comprehensive interpersonal communication competence scale developed by Rubin [20], as modified and supplemented by Hur [21] to fit Korean culture. This tool consists of 15 items related to communication, such as self-exposure, cross-exposure, social tension relief, assertiveness, and concentration. Each item is assessed on a 5-point Likert scale ranging from 1 point, for ‘not at all’, to 5 points, for ‘strongly agree’. The Cronbach’ α reliability measure at the time of development was 0.72, and the value in this study was 0.84.

Problem-solving ability was assessed via a life-skills measurement tool developed by the Korea Educational Development Institute [22]. This tool considers 5 problem factors (clarification, cause analysis, alternative development, plan and implementation, and performance evaluation) and 9 sub-factors (problem recognition, information collection, analysis ability, divergent thinking, decision making, planning ability, execution and risk taking, evaluation, and feedback), and thus consists of 45 items in total. Each item is assessed on a 5-point Likert scale ranging from 1 point, for ‘very rarely’, to 5 points, for ‘very often’, with higher scores indicating better problem-solving skills. The reliability at the time of tool development was indicated by a Cronbach’s α value of 0.95, whereas in this study, the Cronbach’s α value was 0.90.

### 2.4. Statistical Analysis

Statistical analyses were performed using SPSS (ver. 25) statistical software (IBM). The subjects’ general characteristics, metacognition, communication competence, and problem-solving ability were analyzed by number and percentage, as well as mean and standard deviation. To test the normality of all variables, skewness and kurtosis were assessed. In general, when the absolute value of skewness is less than 2 or the absolute value of kurtosis is less than 7, there are no problems associated with deviations in the variable distributions from normality [23]. In this study, skewness ranged between −0.002 and 0.435, with absolute values less than 2, and kurtosis ranged between −0.204 to 1.580, with absolute values less than 7, thus indicating that the variables satisfied the assumption of univariate normality. Differences in metacognition, communication competence, and problem-solving ability according to the general characteristics of the subjects were analyzed by mean, standard deviation, independent t-test, and one-way ANOVA, followed by Scheffé’s post hoc analysis. The correlation between metacognition, communication competence, and problem-solving ability of the participants was analyzed using Pearson’s correlation coefficient. Factors affecting the subject’s problem-solving ability were analyzed using hierarchical multiple regression.

### 2.5. Ethical Considerations

Before the study was conducted, the research proposal and questionnaire were approved by the Institutional Review Board of Gangneung–Wonju National University (No: GWNUIRB-2022-1). The tools used in the study were used after obtaining the consent of the original author. When explaining the purpose of the study, it was emphasized that the participants had the right to withdraw from the study at any time, that the anonymity and confidentiality of the survey results were guaranteed, and that the study results would not be used for other purposes. Participants were provided with a small gift to motivate participation.

## 3. Results

### 3.1. General Characteristics

Table 1 illustrates that the participants’ mean age was 21.56 ± 1.99. Most participants were women (81.3%).

### 3.2. Scores for Metacognition, Communication Competence, and Problem-Solving Ability

Table 2 illustrates that the average score of the subjects’ metacognition was 3.86 ± 0.47 (out of 5). Among the sub-domains, cognitive strategy showed the highest score of 4.03 ± 0.45, followed by monitoring, with 3.90 ± 0.59. The average for communication competence was 3.92 ± 0.42 points (out of 5 points). The average for problem-solving ability was 3.63 ± 0.35 (out of 5), and among the sub-domains, problem clarification was the highest at 3.83 ± 0.52, and cause analysis was the lowest at 3.28 ± 0.36.

### 3.3. Differences in Problem-Solving Ability According to General Characteristics

The problem-solving ability according to the general characteristics of the subjects was as follows (Table 3): age (F = 4.32, *p* = 0.015), academic level (F = 10.17, *p* < 0.001), interpersonal relationships (F = 9.47, *p* < 0.001), satisfaction with major (F = 3.73, *p* = 0.012), PBL experience (F = 3.73, *p* = 0.012), number of PBL experiences (F = 3.20, *p* = 0.025), and practical experience (F = 2.74, *p* = 0.007). There was a significant difference in problem-solving ability accorfing to the number of training weeks (F = 4.46, *p* = 0.013). Scheffé’s post hoc analysis indicated that participants older than 23 years old and younger than 20 years old, as well as fourth-year students, were more dissatisfied than were second-year students. In other cases, interpersonal relationships were very good. Additionally, satisfaction with the major was more than satisfactory. Problem-solving ability was statistically significantly higher for those with more than 7 weeks of practice, and there was no case of not having more than 7 weeks of practice.

### 3.4. Relationship beetween Metacognition, Communication Competence, and Problem-Solving Ability

There was a strong, significantly positive correlation between metacognition and problem-solving ability (r = 0.672, *p* < 0.001), and communication competence and problem-solving ability (r = 0.542, *p* < 0.001). There was also a strong, significantly positive correlation between metacognition and communication competence (r = 0.557, *p* < 0.001; Table 4).

### 3.5. Factors Influencing Problem-Solving Ability

Among general characteristics, variables were converted into dummy variables as needed to confirm their effect on the problem-solving ability of nursing students (e.g., age, 23 years or older = 1; academic level, third year = 1; interpersonal relationships, very good = 1; satisfaction with major, more than satisfied = 1; the number of PBL experiences, 3–6 times = 1; and the number of training weeks, 7 weeks or more = 1). A hierarchical stepwise multiple regression analysis was then performed, inputting communication ability followed by metacognition, which were significantly correlated.

The Durbin–Watson value was 1.96 (close to 2), which confirmed that there was no autocorrelation between the independent variables. The variance inflation factor was 1.013 to 4.999; as all value were less than 10, there were no problems with multicollinearity between independent variables.

General characteristics that showed a significant difference with problem-solving ability in univariate analyses were first input to Model 1, namely age, academic level, interpersonal relationships, and satisfaction with major. This model explained 8.1% of the variance in problem-solving ability. When PBL experience and frequency, clinical practice, and number of weeks were added to Model 1 (Model 2), the variance explained was 12.5%, namely an increased of 4.4% compared to Model 1. When communication competence and metacognition were additionally added to Model 2 (Model 3), the variance explained was 51.2%, which is an increase of 38.7% compared to Model 2. Finally, age (β = 0.11, *p* = 0.048), communication competence (β = 0.24, *p* = 0.001), and metacognition (β = 0.52, *p* = 0.023) were significantly related to problem-solving ability. These variables exhibited an explanatory power of 51.2% (F = 21.01, *p* < 0.001) regarding job satisfaction; the most influential variable was metacognition (Table 5).

## 4. Discussion

In this study, the mean problem-solving ability score of nursing students was 3.63 out of 5, which is similar to the 3.56 points reported in a study targeting third- and fourth- year students in the department of nursing [24]. However, the current value is higher than the 3.44 points reported in a study targeting first- and second-year students [13]. Participants in these studies were nursing students in the second, third, and fourth years of study in this paper, in the third and fourth years in the study done by Kim et al. [24]; and in the first and second grades in the study by Ji et al. [13]. Problem-solving ability can be developed under the influence of various factors; those identified in previous studies include communication ability [13], critical thinking ability [12,25], metacognition [11], and self-directed learning [2]. These factors are continuously improved through various interpersonal relationships formed while learning and studying liberal arts and other major subjects, rather than existing as innate abilities [26]. Therefore, the degree of problem-solving ability was rather high in the study targeting the upper grades. Problem-solving ability in various unexpected situations is essential for working as a nurse [27]. In the current study, the problem-solving ability score of nursing students approximated the 72.6 percentile of the full 100-point scale. Although this score is relatively high, it is nevertheless necessary to improve problem-solving ability; given the nature of the nurse’s job, this ability represents a very important competency. Therefore, it is necessary to improve the problem-solving ability level of nursing students in Korea. The results of this study showed that factors affecting nursing students’ problem-solving ability were metacognition, communication competence, and age. It is necessary to establish a strategy that considers these factors to improve the problem-solving ability of nursing students.

The participants’ average metacognitive score was 3.86 out of 5, which was slightly higher than the 3.61 observed in a study conducted using the same tools for second-year nursing students [12]. While direct comparison using other tools is difficult, the metacognitive level of 72.3 obtained by Kim [28] for all grades in the nursing department was lower than the 77.2 points (out of 100 points) obtained in this study. According to Sternberg and Sternberg [29], the problem-solving phase includes problem identification, problem expression, strategy formulation, information construction, resource allocation, supervision, and evaluation. For health science students, metacognitive instruction has been shown to have a positive effect on students’ problem-solving ability and in improving academic achievement [30]. That is, metacognition is a key factor in predicting learning outcomes in the problem-solving domain [31]. These results were replicated in this study, which showed that a higher metacognitive level of nursing students indicated a significantly higher problem-solving ability. The subjects of this study were second-, third-, and fourth-year nursing students, and it is thought that their metacognitive level was improved compared to students in earlier phases of education, as a result of the curriculum of the nursing department. Educational programs and strategies to improve metacognition will be needed to improve the problem-solving ability of nursing students. The components of metacognition are thinking deeply in the planning stage, establishing possible strategies, undertaking regulating and monitoring activities to carry out the strategy, and revising and regulating to ensure that the solution is progressing in an appropriate direction to achieve the goal [32]. Therefore, a professor who instructs and checks nursing students is necessary to enable them to set their own goals in the curricular and non-curricular programs as they advance through the course, plan and implement strategies to achieve their goals through deep thinking, and conduct their own monitoring and control processes.

The average communication competence score of the subjects was 3.92 out of 5. Previous studies targeting students in various years of the course at the department of nursing found that the communication competence of nursing students also improved as they progressed through the course, with values reported of 3.58 points [33] and 3.56 points [34].

In previous studies, communication competence was a factor affecting nursing students’ problem-solving ability [13]. However, the current study is valuable because it additionally revealed that the level of problem-solving ability significantly increased according to the level of communication competence. Case-based education is suggested as a strategy to simultaneously improve communication competence and problem-solving skills in nursing student education [17]. This is because it is difficult to solve problems through integrative thinking and effective communication in a clinical environment, such as a hospital, with only theoretical knowledge of nursing subjects. Thus, the PBL method is applied to theoretical education in nursing colleges [35,36]. The results of this study showed that the presence or absence of PBL education had a significant effect on the problem-solving ability of nursing students; this ability improved when the PBL factor was added to Model 2. Therefore, it is necessary for nursing professors to practice and improve the communication competence of their students through case-based education in various subjects beginning in the first year to improve the problem-solving abilities of the students.

Additionally, the age of the subjects was also a factor influencing the problem-solving ability. Stewart, Cooper, and Moulding [37] reported that metacognitive levels increase with age. The study revealed that the communication competence of nursing students improved through various experiences [38]. Age may have had a similar influence. 

Previous studies identified critical thinking disposition, empathy, nursing professional intuition, self-leadership [24], learning motivation [17], and communication competence [13] as examples of factors that affect nursing students’ problem-solving ability. However, this study demonstrated that metacognition also significantly affects the problem-solving ability of nursing students. In particular, metacognition and communication competence are considered key concepts, as they explained 51.2% of nursing students’ problem-solving ability. Therefore, it is necessary to consider and continuously apply educational strategies to improve metacognition and communication competence in the education of nursing students in the future.

This study is limited by the small number of nursing colleges that were included in the sampling, as this hinders the generalizability of the results.

## 5. Conclusions

This was a descriptive research study that identified the degree of and correlations between metacognition, communication competence, and problem-solving ability of nursing students, and identified factors that affect problem-solving ability. The results demonstrated that age, communication competence, and metacognition were the factors that most significantly affected the problem-solving ability of nursing students. Among these factors, metacognition had the greatest influence. Therefore, to improve the problem-solving ability of nursing students, an educational strategy is needed to improve communication competence through case-based learning in the curriculum, and development and application of activities such as PBL. In addition, the guidance of professors is needed to enable nursing students to improve their metacognition.

Since this study revealed that metacognition is a factor that influences the problem-solving ability of nursing students, we recommend conducting a study to check whether it affects actual problem-solving by developing and applying a metacognitive improvement curriculum in the future.

## Figures and Tables

**Table 1 healthcare-10-01184-t001:** General Characteristics of Participants (N = 192).

Variable	Categories	N	%
Sex	Male	36	18.7
	Female	156	81.3
Age (years)	<21	62	32.3
	21~<23	84	43.8
	≥23	46	24.0
	Mean ± SD ^†^	21.56 ± 1.99	
Academic level	Sophomore	63	32.8
	Junior	64	33.3
	Senior	65	33.9
Interpersonal relationships	Very good	29	15.1
	Good	118	61.5
	Moderate	45	23.4
Satisfaction with major	Very satisfied	30	15.6
	Satisfied	100	52.1
	Moderately satisfied	47	24.5
	Dissatisfied	15	7.8
PBL ^‡^ experience	Yes	130	68.2
	No	62	31.8
Number of PBL ^‡^ experiences	0	62	32.3
	1~2	29	15.1
	3~6	55	28.6
	≥7	46	24.0
Clinical practice experience	Yes	104	50.2
	No	88	46.8
Weeks of clinical practice experience	0	88	45.9
	1~6	54	28.1
	≥7	50	26.0

^†^ SD, standard deviation; ^‡^ PBL, problem-based learning.

**Table 2 healthcare-10-01184-t002:** Scores for metacognition, communication competence, and problem-solving ability. (N = 192).

Variables	Categories	M ± SD	Range	Min	Max	Skewness	Kurtosis
**Metacognition**	Cognition	3.87 ± 0.54	1–5	2.00	5.00	−0.405	0.616
	Cognitive strategy	4.03 ± 0.45	1–5	2.80	5.00	−0.002	−0.204
	Planning	3.66 ± 0.59	1–5	2.00	5.00	−0.234	0.449
	Monitoring	3.90 ± 0.59	1–5	2.20	5.00	−0.366	0.364
	**Total**	3.86 ± 0.47	1–5	2.35	5.00	−0.214	0.607
**Communication competence**		3.92 ± 0.42	1–5	2.60	4.93	0.025	0.582
**Problem-solving ability**	Problem clarification	3.83 ± 0.52	1–5	1.80	5.00	−0.508	1.288
	Cause analysis	3.28 ± 0.36	1–5	2.50	5.00	0.435	1.171
	Alternative development	3.64 ± 0.47	1–5	2.20	5.00	0.285	0.989
	Planning/implementation	3.63 ± 0.51	1–5	1.50	5.00	−0.203	1.580
	Performance evaluation	3.65 ± 0.39	1–5	2.60	5.00	0.351	1.205
	**Total**	3.63 ± 0.35	1–5	2.53	4.64	0.243	1.253

**Table 3 healthcare-10-01184-t003:** Differences in metacognition, communication competence, and problem-solving ability according to nursing students’ general characteristics (N = 192).

**Variable**	**Categories**	**Metacognition**		**Communication Competence**		**Problem-Solving Ability**	
		M ± SD	t/F(*p*)	M ± SD	t/F(*p*)	M ± SD	t/F(*p*)
Age (years)	<21 ^a^	3.75 ± 0.48	2.83 (0.061)	3.87 ± 0.38	0.84 (0.433)	3.54 ± 0.32	4.32 (0.015) a < c ^†^
	21–<23 ^b^	3.92 ± 0.47		3.95 ± 0.42		3.64 ± 0.37	
	≥23 ^c^	3.92 ± 0.44		3.96 ± 0.47		3.73 ± 0.31	
Sex	Male	3.92 ± 0.42	0.68 (0.409)	3.95 ± 0.41	0.30 (0.584)	3.70 ± 0.35	1.81 (0.181)
	Female	3.85 ± 0.49		3.92 ± 0.42		3.61 ± 0.34	
Academic level	Sophomore ^a^	3.72 ± 0.59	5.59 (0.004) a < c ^†^	3.80 ± 0.43	4.58 (0.011) a < c ^†^	3.51 ± 0.29	10.17 (<0.001) a < c ^†^
	Junior ^b^	3.89 ± 0.47		3.96 ± 0.35		3.60 ± 0.32	
	Senior ^c^	3.99 ± 0.45		4.01 ± 0.44		3.77 ± 0.37	
Interpersonal relationships	Very good ^a^	4.07 ± 0.54	5.79 (0.004) a > c ^†^	4.38 ± 0.40	40.71 (<0.001) c < b < a ^†^	3.78 ± 0.48	9.47 (<0.001) b, c < a ^†^
	Good ^b^	3.88 ± 0.44		3.92 ± 0.32		3.66 ± 0.29	
	Moderate ^c^	3.70 ± 0.48		3.63 ± 0.39		3.46 ± 0.33	
Satisfaction with major	Very satisfied ^a^	4.09 ± 0.43	7.21 (<0.001) d < b, c < a ^†^	4.19 ± 0.48	7.70 (<0.001) c, d < a ^†^	3.76 ± 0.38	3.73 (0.012) d < a, b ^†^
	Satisfied ^b^	3.92 ± 0.45		3.94 ± 0.38		3.66 ± 0.33	
	Moderately satisfied ^c^	3.71 ± 0.50		3.77 ± 0.40		3.53 ± 0.35	
	Dissatisfied ^d^	3.55 ± 0.37		3.77 ± 0.31		3.52 ± 0.23	
PBL ^‡^ experience	Yes	3.93 ± 0.47	2.92 (0.004)	3.98 ± 0.40	2.74 (0.007)	3.68 ± 0.36	3.06 (0.003)
	No	3.72 ± 0.47		3.81 ± 0.43		3.52 ± 0.28	
Number of PBL ^‡^ experiences	0	3.72 ± 0.47	2.87 (0.038)	3.81 ± 0.43	2.47 (0.063)	3.52 ± 0.28	3.20 (0.025)
	1~2	3.92 ± 0.51		3.97 ± 0.47		3.66 ± 0.42	
	3~6	3.95 ± 0.47		3.98 ± 0.38		3.70 ± 0.36	
	≥7	3.91 ± 0.44		3.98 ± 0.38		3.68 ± 0.32	
Clinical practice experience	Yes	3.93 ± 0.42	2.40 (0.019)	3.97 ± 0.38	1.58 (0.115)	3.69 ± 0.32	2.74 (0.007)
	No	3.78 ± 0.52		3.91 ± 0.44		3.56 ± 0.36	
Weeks of clinical practice experience	0	3.78 ± 0.52	3.02 (0.051)	3.87 ± 0.46	1.58 (0.208)	3.56 ± 0.36	4.46 (0.013) a < c ^†^
	1~6	3.96 ± 0.40		3.93 ± 0.35		3.66 ± 0.29	
	≥7	3.91 ± 0.44		4.00 ± 0.41		3.73 ± 0.35	

^†^ Scheffé test; ^‡^ PBL, problem based learning.

**Table 4 healthcare-10-01184-t004:** Relationships between metacognition, communication competence, and problem-solving ability (N = 192).

Variables	Metacognition	Communication Competence	Problem-Solving Ability
	r (*p*)	r (*p*)	r (*p*)
Metacognition	1		
Communication competence	0.557 (<0.001)	1	
Problem-solving ability	0.672 (<0.001)	0.542 (<0.001)	1

**Table 5 healthcare-10-01184-t005:** Factors influencing problem-solving ability (N = 192).

Variables	Model 1			Model 2			Model 3		
	B	β	t	B	β	t	B	β	t
Age ^†^ (R ^‡^ = ≥23)	0.13	0.16	2.29 *	0.10	0.13	1.77	0.09	0.11	1.99 *
Academic level ^†^ (R ^‡^ = Junior)	−0.06	−0.09	−1.22	−0.23	−0.31	−2.71 *	−0.12	−0.17	1.91
Interpersonal relationships ^†^ (R ^‡^ = Very good)	0.16	0.17	2.41 *	0.14	0.15	2.11 *	−0.19	−0.02	−0.33
Satisfaction with major ^†^ (R ^‡^ = Satisfied)	0.13	0.17	2.40 *	0.11	0.15	2.10 *	−0.01	−0.02	−0.35
PBL ^¶^ experience				−0.25	−0.34	−2.25 *	−0.09	−0.11	−1.01
Number of PBL ^¶^ experiences ^†^ (R ^‡^ = 3–6)				0.14	0.19	1.66	0.14	0.18	2.13 *
Clinical practice experience				0.11	0.16	1.16	0.11	0.16	1.16
Weeks of clinical practice experience ^†^ (R ^‡^ = ≥7)				−0.00	−0.004	−0.03	0.09	0.12	−0.03
Communication competence							0.19	0.24	3.42 *
Metacognition							0.38	0.52	8.22 *
	Adj R^2^ = 0.081, F = 5.20, *p* = 0.001			Adj R^2^ = 0.125, F = 4.42, *p* < 0.001			Adj R^2^ = 0.512, F = 21.01, *p* < 0.001		

^†^ Dummy variables; ^‡^ Reference; ^¶^ PBL, problem-based learning; * *p* < 0.05.

## Data Availability

Not applicable.

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
