# Peer review of "Factors Influencing Problem-Solving Competence of Nursing Students: A Cross-Sectional Study"

_healthcare, 2022, doi:10.3390/healthcare10071184_

Round 1
Reviewer 1 Report
Dear author(s),
It was a pleasure for me to read your article. Especially the results you find are scientifically valuable. For this reason, I congratulate you. However, there are issues that need to be improved in various parts of the article. I think doing this will increase the quality of your article. My recommendations are presented below.
I wish you good work.
Kind regards.
In the article, I suggest developing the following aspects:
1. The abstract should state where the research was conducted and the method.
2. In the introduction, a paragraph should be written right after the first paragraph of this research, and the purpose, importance and subject of this research should be stated. The following researches can be used both in the background and in the discussion section:
• A path model for metacognition and its relation to problem-solving strategies and achievement for different tasks http://dx.doi.org.ezproxy.neu.edu.tr:2048/10.1007/s11858-019-01067-3
• The Mediator Role of Critical Thinking Disposition in the Relationship between Perceived Problem-Solving Skills and Metacognitive Awareness of Gifted and Talented Students. https://doi.org/10.17275/per.22.4.9.1
3. At the end of the Background section, it should be stated more convincingly which gap the research will fill in the literature.
4. Although the research was carried out with quantitative method in the method section, it was not mentioned at all. More details should be given about the design of the study. In the method section, both the research model and the reason why this model was chosen should be mentioned.
It is not specified where the research was conducted.
How big is the universe? Since the universe is not specified, the minimum number of samples hangs. In other words, since no information is given about the universe, the power of 192 participants to represent the universe is not clear. This section should be corrected.
“Major satisfaction” is not clear. Does satisfaction here mean professional satisfaction? If so, it should be stated more clearly. However, this may introduce another problem. Namely; satisfaction refers to an attitude and it is unclear how this was measured in the research.
The results of the normality analysis, which is the first step in the analysis of the data, are not mentioned. Normality test results should be explained.
By opening a new column at the end of each row in the tables, it should also be stated what the average value in this row means. Is this score a high value, a medium value, or a low value?
In addition, the meaning of the point values throughout the article remained suspended. For example, what does the value 3.63 correspond to, high, medium or low? This is not disclosed. It should be interpreted in accordance with the Likert structure.
In correlation analysis, the scales can be considered as a whole, but looking at the relationships between the sub-dimensions will provide a more in-depth analysis. Because each scale has sub-dimensions. This part is missing. In addition, it was stated that there was a positive and significant relationship between the scales, but -just as the meaning of the above point values was not given- the strength of this relationship was not expressed. High, medium or low?
5. In the discussion section, the result of this research is given and compared with the results of two separate studies. It's nice so far. However, what is missing; what is the reason for the result of this research? Why is it 3.63 in this study? What could be the reason for this? Why is it similar to or different from other studies? The main thing to do in the discussion section is to give the research result, compare it with the information in the literature and comment on why this result came out. In addition, the references given to the studies in the literature in the discussion section are not sufficient. Comparisons with studies in the literature should be made in more depth.
Reviewer 2 Report
When metacognition first occurs in the paper, there should be a simple explanation as to what metacognition means .
Reviewer 3 Report
The most relevant aspects of the article, such as the research design and the conclusions, must be clarified extensively.
Reviewer 4 Report
This is a very interesting topic and well described. Please improve on the following points.
Lines 89-93:
The text highlighted in yellow here should be moved to "2.4. Statistical Analysis."
Also, only the sample size for multiple regression analysis has been calculated. Authors must indicate the sample size needed for the other statistical treatments (independent t-test, and one-way ANOVA, and Pearson's correlation coefficient).
Lines 110-118:
by [17].... .by [20],
I think it should be expressed as follows.
by O'Neil Jr et al. [17].
Please refer to some comments in the PDF.
I hope this information will be of assistance to you.
Thank you very much.

Round 2
Reviewer 1 Report
Dear authors,
I think your corrections are on point and good. You did a good job. Congratulations.
Kind regards.